# Skin Lesions with Loss of Tissue and Cutaneous-Onset Sepsis: The Skin Infection–Sepsis Relationship

**DOI:** 10.3390/diagnostics14060659

**Published:** 2024-03-21

**Authors:** Adriana-Ionela Pătrașcu, Dan Vâță, Doinița Temelie-Olinici, Mădălina Mocanu, Dumitrița-Lenuța Guguluș, Mădălina Marinescu, Liviu Stafie, Bogdan-Marian Tarcău, Ioana Creţu, Ioana-Adriana Popescu, Carmen-Diana Cimpoeșu, Laura Gheucă-Solovăstru

**Affiliations:** 1Clinic of Dermatology-Venereology, “Saint Spiridon” Emergency County Clinical Hospital, 700111 Iași, Romania; doinitzaganceanu@yahoo.com (D.T.-O.); drmadalinamocanu@yahoo.com (M.M.); bogdan.tarcau@yahoo.com (B.-M.T.); oana.manolache@yahoo.com (I.-A.P.); lsolovastru13@yahoo.com (L.G.-S.); 2Discipline of Dermatology-Venereology, Department of Medical Sciences III, Faculty of Medicine “Grigore T. Popa”, University of Medicine and Pharmacy, 700115 Iași, Romania; nichitean.dumi@yahoo.com (D.-L.G.); dr.madalinamarinescu@gmail.com (M.M.); 3Discipline of Cellular and Molecular Biology, Department of Morpho-Functional Sciences II, Faculty of Medicine “Grigore T. Popa”, University of Medicine and Pharmacy, 700115 Iași, Romania; 4Clinic of Diabetes Mellitus and Nutrition, “Sf. Spiridon” Emergency County Clinical Hospital, 700111 Iași, Romania; 5Discipline of Public Health and Management, Department of Preventive Medicine and Interdisciplinarity, Faculty of Medicine “Grigore T. Popa”, University of Medicine and Pharmacy, 700115 Iași, Romania; dr_liviustafie@yahoo.com; 6Crețu R. Ioana Persoană Fizică Autorizată, 707020 Aroneanu, Romania; contact@ioanacretu.ro; 7Discipline of Emergency, Medicine Department of Surgery II, Faculty of Medicine “Grigore T. Popa”, University of Medicine and Pharmacy, 700115 Iași, Romania; carmen.cimpoesu@umfiasi.ro

**Keywords:** skin lesions with loss of tissue, cutaneous-onset sepsis, acute infections, skin microbiome

## Abstract

Infectious and inflammatory dermatoses featuring skin lesions with loss of tissue expose skin layers to microbial invasions, disrupt the normal skin microbiome, and potentially lead to sepsis. However, literature data on the incidence of cutaneous-onset sepsis are scarce. This retrospective observational study assessed hospital admissions for primary skin lesions without bacterial infections and sepsis during 2020–2022 in the largest emergency hospital in NE Romania. Of 509 patients, 441 had infected lesions, 78 had sepsis caused by venous ulcers from microbial eczema cellulitis, superinfected bullous dermatoses, erysipelas, and erythroderma. Cultured samples revealed *S. aureus*, *P. aeruginosa*, and *E. coli*; and *K. pneumoniae* and *S. β-hemolytic* associated with sepsis, even if this was rarer. Clinical manifestations included ulcerations, erosions, fissures, excoriations, bullae, vesicles, pruritus, tumefaction, edema, fever, chills, pain, adenopathy, and mildly altered mental status. Underlying chronic heart failure, atrial fibrillation, anemia, and type-1 diabetes mellitus were comorbidities associated with infection and sepsis. Significant associations and risk factors, including their combined effects, are discussed to draw attention to the need for further research and adequate management to prevent sepsis in adult patients of any age presenting with infected skin lesions (especially cellulitis) and comorbidities (especially type 1 diabetes mellitus and anemia).

## 1. Introduction

Acute skin infections, regardless of etiology (bacterial, viral, or fungal), are very frequent in dermatological practice. Whether superficial or deep, mild or severe, they can aggravate further and lead to sepsis, triggering a complex cascade of dysfunction and even failure across multiple organs and body systems. Addressing the first clinical signs of skin infections promptly and effectively is the most appropriate therapeutic approach [1].

Sepsis of any type is a serious threat to public health, and the international research community seeks to establish sepsis definitions, treatments, and preventive measures [2]. The understanding of sepsis has undergone reconceptualization over time. In 1991, the American College of Chest Physicians described *sepsis* as a *systemic inflammatory response syndrome* (SIRS) caused by infection, with possible further aggravation to *severe sepsis* in case of organ dysfunction with hypoperfusion or hypotension; and to *septic shock* in case of hypotension despite adequate volemic resuscitation. In 2016, the Third International Consensus Definitions for Sepsis and Septic Shock defined *sepsis-3* as life-threatening organ dysfunction, acknowledging the severe, lethal threat of a pathogen invading the body [3,4]. The diagnosis of organ dysfunction is now based on Sequential Organ Failure Assessment (SOFA) score changes of at least two points as a result of acute infection [5].

According to public health data, the sepsis-related mortality rate was 41% in Europe and 28.3% in the United States in 2012 [6]. In 2017, 48.9 million cases of sepsis were reported worldwide, resulting in 11 million deaths, or 19.7% of all deaths that year [7]. These numbers are still cited, suggesting a need for more updated epidemiological reports.

The starting point of infection in sepsis patients has been described, but few studies highlight the importance of skin semiology in the early recognition of sepsis. One such study conducted in Colombia on a group of patients diagnosed with sepsis showed that skin and soft tissue infections were the fourth leading cause of sepsis (9.4%) after urinary tract infections (27.6%), lower respiratory tract infections (27.4%), and intra-abdominal infections (10.8%). Almost a quarter of sepsis episodes originating in skin infections progressed to septic shock, and 28% of these patients had to be transferred to intensive care units. Of them, 8% succumbed to their conditions [8].

The skin is home to an incredibly diverse microbiome, normally maintained in a healthy balance. When the skin’s defense mechanisms are overrun by proliferating pathogens, the ensuing acute skin infections can be cellulitis, erysipelas, trophic ulcers in advanced stages of chronic venous insufficiency, bullous dermatoses (pemphigus, pemphigoid, erythema multiforme) with extensive denudation, and erythroderma. The bacterial agents most often involved are Methicillin-resistant *Staphylococcus aureus* (MRSA), *Escherichia coli*, *Pseudomonas aeruginosa*, species of the genus *Streptococcus*, *Klebsiella pneumoniae*, *Proteus mirabilis*, and *Enterococcus* [9]. The onset of infectious dermatoses is abrupt, with soft tissue erythema, induration, local heat, pain, fever, followed by the deterioration of vital functions and single/multiple organ dysfunction as sepsis complications.

The appropriate management of acute skin infections depends on their severity. Mild forms can be effectively resolved with topical antibiotics, while severe infections require systemic antibiotic therapy. Initiation should be prompt and with agents empirically proven to be sensitive to gram-positive microorganisms, such as penicillinase-resistant penicillin, cephalosporins, macrolides, or fluoroquinolones. In case of severe sepsis, adequate treatment is multifactorial and multidisciplinary, including the administration of vasopressor agents, steroids, anticoagulants, anti-inflammatories, as well as glycemic control, ventilation support, and even (early) resuscitation [10,11]. Medical teams must consider the infection gateway, the pathogens involved, the patient’s age, and comorbidities. In most cases, a broad-spectrum empiric antibiotic with good tissue penetration is administered first, followed by switching to a narrow-spectrum antibiotic therapy based on the antibiogram results [12].

### Study Aims

In this study, we assessed the incidence of skin lesions with loss of tissue that can become infected and complicated further by cutaneous-onset sepsis, in the context of tertiary care provided by the dermatology unit of the largest emergency hospital in NE Romania. The specific objectives included: (1) surveying hospital records, (2) compiling a comprehensive database on all patients admitted with skin lesions featuring loss of tissue, (3) analyzing the data to identify statistically significant associations, and (4) reporting results internationally to update awareness and understanding of relevant issues.

The study is relevant considering the scarcity of incidence data on cutaneous-onset sepsis in the literature and the implications for multidisciplinary clinical practice and further research. Skin lesions are not typically regarded as emergencies, but lack of timely diagnosis and treatment can lead to infections and then to sepsis, delaying recovery, increasing treatment complexity and financial costs, undermining quality of life, and even threatening the patient’s life.

## 2. Materials and Methods

### 2.1. Study Coordinates; Inclusion and Exclusion Criteria

This was a retrospective observational study of patients admitted to the Clinic of Dermatology-Venereology of the Emergency Clinical County Hospital “Sf. Spiridon” Iași in NE Romania between January 2020 and December 2022. The study was conducted with the formal approval of the institutional Research Ethics Committees of the hospital and the medical university.

The study enrolled adult patients admitted consecutively for the diagnosis and treatment of skin lesions (ulcerations, erosions, fissures, excoriations) and symptoms indicative of associated inflammatory processes (edema, tumefaction, erythema, fever, pain, etc.). Completing a bacteriological examination on admission was a key criterion for inclusion. Patients younger than 18 with concomitant SARS-CoV-2 infection in sepsis of non-cutaneous cause and/or with Glasgow Coma Scores < 13) were excluded from the study.

A detailed clinical analysis of skin manifestations was performed, and the patient data were organized in three study groups:group A—infected skin lesions and sepsis;group B—infected lesions without sepsis;group C—non-infected skin lesions (negative bacteriological examination results).

The three datasets were compared to identify significant associations between clinical manifestations (cutaneous, systemic) and patient characteristics (demographics, comorbidities, mental status), as well as to assess risk factors.

### 2.2. Definitions

The types of skin lesions with loss of tissue considered in this study were: erosions (loss of tissue down to the skin’s basal membrane), ulcerations (deep loss of tissue), fissures (linear lesion-related skin dehydration, thickening, compromised elasticity), excoriations (superficial loss of skin tissue, e.g., due to scratching). The presence of bullae and vesicles was also noted. Necrotizing fasciitis and abscesses could not be included in the analysis due to lack of cases during the studied period.

Sepsis was defined as infection-related systemic inflammatory response syndrome (SIRS) and established when two or more of the following criteria were met: body temperature >38 °C or <36 °C, tachycardia, tachypnea, leukocytosis/leukocytopenia. The differential diagnosis of cutaneous-onset sepsis was based on the identification of primary skin lesions and the confirmation of bacterial agents in cultured samples from the lesions, distinguishing it from sepsis of other causes, e.g., neoplasms, respiratory, urogenital, etc.

Single-use swab tubes with Amies medium were used to collect skin lesion specimens on admission. This type of swab and medium maintains the viability of pathogens during transport. The samples were taken to the hospital’s laboratory within 4 h of being harvested, where they were cultured and tested for sensitivity to antibiotics.

A mild deterioration of mental status on the Glasgow Coma Scale (scores of 13 and 14) was also considered to be indicative of sepsis. This scale is commonly used to assess brain injury, and assigning a GCS score is standard procedure in emergency admissions. Fully awake, responsive, cognitively agile patients receive the maximum score of 15; scores 9–12 describe moderate impairment; and scores of 8 or less indicate severe coma states of unconsciousness. Severely altered mental status occurs in septic shock, which was not the object of our study (patients would not have been referred to the dermatology unit of the hospital; they would have been treated in the ICU of the emergency department instead).

### 2.3. Statistical Analysis

The statistical analysis was performed using the SPSS version 29.0 software package (SPSS Inc., Chicago, IL, USA). Categorical variables were presented as frequencies and percentages, and continuous variables were presented as means ± standard deviation. Categorical variables were compared between groups using the Pearson chi-squared test, and the associated risks (OR) were also calculated. Continuous variables were compared between groups using the Mann–Whitney test (because the pre-condition of normal repartition of values was not verified). To investigate the combined action of the statistically significant risk factors for infection and sepsis, the multivariate analysis was performed using a model for binary logistic regression. Statistical significance was assessed relative to the threshold of *p* < 0.05, and the confidence interval (CI) was set at 95%.

## 3. Results

### 3.1. General Demographic Characteristics

During the three-year research period, 509 admitted patients were diagnosed with infectious dermatoses and met the criteria for inclusion. The patients were between 18 and 92 years old (mean age: 64.22 ± 14.699). In 441 cases, lesions were infected, and sepsis was confirmed in 78 of these cases (group A). Patients with sepsis were older (mean age: 65.23 ± 14.056), but not significantly so (weak statistical significance, *p* = 0.046). These age characteristics are summarized in Table 1.

The 441 patients with infected lesions and sepsis were mostly men (56.2%) and mostly rural residents (57.4%) (see Table 2). Statistically non-significant differences were noted relative to the presence or absence of sepsis, but sex and background differences were significant within each group. Namely, acute skin infections were more common in men than in women (57.0% vs. 43%) and in patients from rural rather than urban areas (58.1% vs. 41.9%). Similarly, sepsis was more common in men (52.6%) and rural residents (53.8%). Male patients appeared to be 2.282 times more at risk of skin infections than women, while urban residence seemed to provide some protection, considering the 0.474-fold higher risk associated with rural background.

### 3.2. Clinical Manifestations—Cutaneous and Systemic, Including Conscious State

Overall, the main clinical manifestations recorded were skin ulcerations, erosions, fissures, excoriations, bullae, vesicles, pruritus, tumefaction, as well as fever, chills, pain, edema, and adenopathy (see Figure 1, Table 3 and Table 4).

The topography of skin lesions was consistent across groups, with non-significant differences between patients with sepsis versus those with uncomplicated infections. Most skin manifestations occurred in the lower limbs (97.4% in group A, 93.9% in group B), and a minority of patients also had lesions on their torsos (23.1% in both groups), upper limbs (17.9% in group A, 18.5% in group B), and faces (10.3% in group A, 8.8% in group B). The location of the studied lesions was not a significant factor in the patients’ aggravation towards sepsis. The same can be said about the number of lesions: patients with multiple lesions were not more likely to progress to sepsis compared to those presenting a single lesion.

While most patients across groups had skin ulcerations, erosions, fissures, and excoriations, significantly fewer patients with sepsis had such lesions (89.7%) compared to patients without complicated infections (97.8%, *p* = 0.003) or those without infections altogether (95.6%). Similarly, bullae or vesicles were least present in patients with sepsis (only 21.8% compared to 29.2% in group B, *p* < 0.001), and pruritus occurred in only 29.5% of patients with sepsis compared to more than 60% of patients in groups B and C (*p* < 0.001). Tumefaction was the only exception, occurring in a majority of cases in all three groups, but mostly in sepsis (*p* = 0.016).

The opposite could be seen in manifestations reaching beyond the skin, such as edema. Relative to the 80.8% of patients with sepsis who manifested edema, significantly fewer patients with uncomplicated infections had it (59.2%), and only 30.9% of uninfected patients did (*p* < 0.001). This confirms edema as a risk factor for both infection (3.251) and sepsis (2.891).

In addition, patients with sepsis experienced systemic signs and symptoms significantly more than the other patients. Fever and chills were, by far, most common in patients with sepsis (80.8%) and were much rarer in the other groups (22.3% and 20.6%, respectively, *p* < 0.001). This translates to a 14.622-fold increased risk of sepsis in the presence of fever, which is not surprising considering what sepsis entails.

Similarly, pain was reported by almost all the patients with sepsis and infection (96.2% and 89.0%, respectively), compared to 61.8% of uninfected patients (*p* < 0.001). The risk analysis pointed to pain as the second highest risk factor for infection (4.999), suggesting that painful symptoms can betray infective aggravation even more than edema, an indicative sign of inflammation.

Regarding adenopathy, even if it occurred in only 17.9% of sepsis cases, it was significantly rarer in the group with uncomplicated infections (5.2%, *p* < 0.001). A 3.961-fold increased risk of developing sepsis in the presence of adenopathy was calculated, but note that adenopathy also occurred in 10.3% of uninfected patients (*p* = 0.158).

Furthermore, a mildly altered state of consciousness (Glasgow scores of 13 or 14) was significantly associated with the presence of sepsis. Of the 13 patients with such scores, most had confirmed sepsis (9% relative to the size of group A vs. 1.7% in group B, *p* = 0.003, see Table 5). The risk assessment associated Glasgow scores of 13 or 14 with a 5.866-fold higher risk of sepsis. This relationship between sepsis and mildly impaired brain function highlights the importance of addressing and preventing sepsis not just generally, but also in the context of treating skin lesions with infective complications.

### 3.3. Main Acute Skin Conditions and the Etiological Agents Responsible for Infections

As summarized in Table 6, the types of diagnosed skin infections were venous ulcers (68.3%), microbial eczema (61.9%), cellulitis (21.1%), superinfected bullous dermatosis such as pemphigus, pemphigoid, Stevens–Johnson syndrome (6.6%), erysipelas (5%), and erythroderma (2%). Other dermatoses featuring loss of tissue or the disruption of the skin barrier, such as atopic dermatitis, psoriasis, vasculitis, and ulcerated skin neoplasms, amounted to 27%.

While venous ulcers were almost equally present in groups A and B (66.7% and 68.6%, respectively), they were by far the least common in uninfected patients (13.2%, *p* < 0.001), see Table 7. Concurrently, microbial eczema was significantly more frequent among patients with uncomplicated infections (68.9%) than both sepsis patients (29.5%) and uninfected patients (27.9%) (*p* < 0.001). However, 48.7% of patients with sepsis had cellulitis, while only 15.2% of patients with uncomplicated infections and just one uninfected patient did (*p* < 0.001). Erysipelas was infrequent in all groups, but rarest among patients with uncomplicated infections (3.9% in group B vs. 10.3% in group A, *p* = 0.038, and 17.6% in group C, *p* < 0.001, respectively). Overall, cellulitis presented the highest risk for sepsis (5.320), followed by erythroderma (3.870) and erysipelas (2.849). Other differences in the incidence of venous ulcers, erythroderma, superinfected bullous dermatosis, and other dermatoses disruptive of the skin barrier were not significant.

The bacterial agents evidenced in the cultured samples from the 441 patients with uncomplicated infections and with sepsis were *Staphylococcus aureus* (43.3%), *Pseudomonas aeruginosa* (33.8%), *Escherichia coli* (12.9%), *Klebsiella pneumoniae* (9.8%), and group A *Streptococcus β-hemolytic* (5.9%). As can be seen in Table 8, only *K. pneumoniae* and *S. β-hemolytic* were significantly more common in the sepsis group (17.9% vs. 8%, *p* = 0.007, and 11.5% vs. 4.7%, *p* = 0.031, respectively). The noted differences contributed to a risk of sepsis that was 2.655 times higher in streptococcal infections and a 2.519-fold increase in the presence of *K. pneumoniae*.

In addition, various other species of bacteria were detected (30.4%) without apparent significant impact on sepsis (*Proteus mirabilis*, *Serratia marcescens*, *Citrobacter freundii*, *Providencia stuartii*, *Stenotrophomonas maltophilia*, *Morganella morganii*, and *Enterobacter* genus). The same can be said about the viral agents found in 9.5% of cases. The hepatitis B and C viruses identified were inactive and did not play an active role in the studied pathologies.

### 3.4. Comorbidities

The associated pathologies of the patients enrolled in the study included cardiovascular diseases (hypertension/HT, chronic heart failure/CHF, atrial fibrillation/AF, obliterating arteriopathy of the lower limbs/OALL), microcytic hypochromic anemia, diabetes mellitus types 1 and 2 (T1DM and T2DM); and neurological, renal/genital, and respiratory conditions summarized in Figure 2 and in Table 9 and Table 10.

Overall, the most common underlying conditions were cardiovascular, of which hypertension was prevalent in all groups, with a non-significant 10% difference between patients with sepsis versus uninfected lesions. Established chronic heart failure was significantly more frequent among patients with sepsis and those with uncomplicated infections (compared to uninfected patients). The risk of infection was 1.924 times higher in the presence of stage I and II CHF according to the classification of the New York Heart Association (NYHA). The same can be said about established atrial fibrillation, which increased the risk of infection 2.527-fold according to our analysis. Lower limb arteriopathy was the least common, and related differences were not conclusive.

Microcytic hypochromic anemia stood out as significantly more common in the sepsis group (55.1% vs. 37.2%, *p* = 0.003). A 2.075-fold increase in the risk of sepsis was thus calculated for patients with anemia, which suggests the importance of timely, adequate treatment of both the underlying hematological condition and of any incipient infection of skin lesions in patients with anemia.

Regarding diabetes mellitus, type 1 was significantly associated with sepsis, although patient numbers were too small to make a reliable determination (only seven cases). This does not mean that T1DM patients are unlikely to develop skin conditions with related infections or sepsis, but rather that it is possible that such patients were prioritized for emergency or diabetes treatment (or treatment in another department). The fact that all T1DM patients admitted in our unit had infected lesions or even sepsis is suggestive of the risks associated with T1DM. The 6.486-fold increased risk among these patients invites further targeted research of the role of underlying T1DM in the progression to sepsis of skin lesion infections. By contrast, type 2 diabetes was more common (104 vs. 7 patients overall), and it was noted in all three study groups. Even if it was most frequent in patients with sepsis (25.6%) and with infected lesions (20.1%), and less so in uninfected patients (16.2%), these differences did not constitute statistical significance.

Concurrently, neurological conditions were significantly more common in patients with sepsis (32.1%) and infected lesions (29.8%) than in the uninfected group (16.2%). Neurological health plays a role in the body’s ability to heal and to fight off pathogens, as this result appears to illustrate this with regard to the skin barrier. The distribution of other respiratory or renal/genital conditions noted in our patients was not associated with our patients’ status relative to infection and sepsis.

### 3.5. Multivariate Logistic Regression Analyses of Risk Factors for Infection and Sepsis

Six-step logistic regression models were built to assess the combined risk effects of parameters previously identified as significantly associated with infection and, respectively, with sepsis.

The model generated to assess the combined risk for infection explained 42.5% of the variations recorded in the incidence of infected skin lesions and adequately classified 87.2% of cases with 97.5% sensitivity and 32.4% specificity (*p* < 0.001). The Nagelkerke R^2^ coefficient was used to determine to what extent infection could be explained by the variables of interest (the risk factors flagged as statistically significant in Table 11).

The cumulated statistical effect of various combinations of these risk factors was also analyzed to establish the probability of infection (absent—0/present—1, relevant risk > 0.5). As can be seen in Table 12, the risk of infection was highest (especially with *E. coli*) in rural residents with histories of neurological conditions who were admitted with venous ulcers, cellulitis, and microbial eczema. The risk remained at the highest level even in the absence of one or two of these factors.

The model generated to assess the risk of sepsis explained 41.9% of the variations recorded in the incidence of sepsis (the Nagelkerke R^2^ coefficient) and correctly classified 84.8% of cases with 38.5% specificity and 94.8% sensitivity (*p* < 0.001). The most significant risk factors for sepsis were different to those for the risk of infection: mildly altered mental status, fever and chills, adenopathy, and established histories of type 1 diabetes mellitus and anemia. Notably, only the diagnosis of cellulitis featured in both models (see Table 13).

The probability of sepsis in the combined presence of significant individual risk factors is explored in Table 14. The risk was highest in patients with underlying type-1 diabetes mellitus and anemia and with lesions diagnosed as cellulitis; and in those admitted with fever, chills, adenopathy, and mildly altered mental status. The risk remained excessively high in the absence of one, two, or even three of these factors. Absent fever and chills brought the risk of sepsis to irrelevant levels, as can be expected.

## 4. Discussion

### 4.1. Study Results Overview and General Considerations

To prevent infection and further complication to sepsis, dermatological conditions disruptive of the skin barrier should be promptly diagnosed and adequately treated. In this study, we analyzed the incidence of skin lesions associated with infections and sepsis in 509 adult patients from NE Romania. The patients were admitted to the dermatology unit of the largest emergency hospital in the region over a three-year period (2020–2022), and those with concomitant infectious conditions or sepsis of other causes, including SARS-CoV-2, were not included. The patients presented lesions with loss of tissue, e.g., ulcerations, erosions, fissures, excoriations, as well as bullae and vesicles. More severe cases of necrotizing fasciitis and abscess were not recorded. The accompanying signs and symptoms ranged from pruritus, erythema, tumefaction to edema, pain, fever, chills, and adenopathy, which are indicative signs of inflammatory processes often caused by infection, but also by mildly altered mental status. Etiological agents, mostly bacterial, were found to superinfect the patients’ skin lesions in 363 cases (71.3%), and 78 patients were in sepsis (15.3%). Lesions were uncontaminated in 68 patients (13.3%), which does not preclude infection and (self-)treatment prior to admission.

Research data on the incidence of acute skin infections and cutaneous-onset sepsis are currently scarce. An analysis of national US care data surveys from 1997 to 2005 revealed a 50% rise in visit rates from increasingly younger patients to diagnose and treat skin and soft tissue infections (SSTIs), especially in emergency settings, and mostly for cellulitis or abscesses [13]. In addition, there were 29% more hospital admissions in the US due to SSTIs in 2004 than in 2000, reaching close to 900,000 acute-care admissions, while the number of pneumonia-related hospitalizations remained roughly the same [14].

In Europe, 11% of infection-related presentations in Spanish emergency departments were SSTIs (1250 cases). Of them, 3.3% had signs and symptoms of septic syndrome [15]. In a recent review, sepsis or bacteremia cases constituted between 4.8% to 16% of SSTIs, with our result notably approaching the higher end of this range [8]. The seven-year SENTRY Antimicrobial Surveillance Program implemented in Europe, North America, and Latin America associated the rising incidence of skin infections across all regions with *Staphylococcus aureus*, *Pseudomonas aeruginosa*, *Escherichia coli*, and *Enterococcus* species [16].

Advanced age is a known risk factor in the progression to sepsis, as elderly patients are more likely to suffer from diminished immune response, malnutrition, multiple comorbidities, poorer body hygiene, skin injuries, etc. These vulnerabilities combined can complicate and accelerate infection towards septic shock and even exitus, as shown by numerous studies. For instance, Gabriel Wardi et al. reported close correlations between the severity of associated pathologies and both extent of organ dysfunction and unfavorable prognosis [17]. However, our analysis of a significant sample of patients of all adult ages showed that age differences made far less of a difference than comorbidities, a key finding in our study.

Information on sepsis epidemiology and patterns in patients under the age of 60 is limited but not altogether absent. Carmen Bouza surveyed the national Spanish health data between 2006 and 2015, finding 28,351 cases of sepsis in patients aged 20–44 years, which amounted to 3.06‰ of all hospitalizations for this age group [18]. Considering the overall age characteristics of our patients (age range 18 to 92, mean age 64.22, median 65), it is worth noting that cutaneous-onset sepsis occurred in patients of all ages (18–88); they were in fact younger on average (mean 61.33, median 62) compared to patients without sepsis, at a weakly significant *p* = 0.046. This result challenges existing notions related to age and sepsis, inviting further research into other potentiating factors such as comorbidities. It also underscores the importance of addressing skin lesions before they become contaminated.

Regarding background, rural residence has generally been proven to limit or delay access to health services. Primary and secondary outpatient or hospital care services are more readily available in towns and cities. Delayed diagnosis and management lead to further aggravation and complications that statistically show up as higher rates of infection, sepsis, and mortality among patients from rural areas [19]. Our data, which are focused exclusively on skin lesions, make this distinction very clear for all three studied situations: sepsis, uncomplicated infections, and uncontaminated lesions.

### 4.2. In-Depth Discussion of Studied Lesions

The onset of infectious dermatoses is often sudden, with soft tissue erythema, induration, local heat, and pain, followed by more generalized symptoms such as fever, deterioration of vital functions, and single or multiple organ dysfunction in the case of septic complications. Careful analysis of local and systemic manifestations facilitates correct diagnosis and prompt initiation of appropriate treatment. The acute skin infections diagnosed in our patients included trophic ulcers in advanced stages of chronic venous insufficiency, microbial eczema, cellulitis, erysipelas, bullous dermatoses (pemphigus, pemphigoid, erythema multiforme, Stevens–Johnson syndrome), erythroderma, and other dermatoses (ulcerated cutaneous neoplasms, atopic dermatitis in the exudative phase, tinea pedis intertriginosa).

Cellulitis typically involves a bacterial infection of the dermis and subcutaneous cellular tissue, most commonly the lower limbs (70–80% of cases) [20]. A recurrent medical emergency, it starts abruptly with soft tissue erythema, warmth, and local tenderness, and is most often caused by *Streptococcus pyogenes* and/or *Staphylococcus aureus*. In our study, more than 90% of the 93 patients with cellulitis had lower limb lesions, and just one patient was not infected. In approximately 40% of cases, multiple types of bacteria were evidenced, the other 60% being monobacterial infections. The main species found were *P. aeruginosa* (38.7%) and *Staphylococcus* (35.48%), while *K. pneumoniae*, *E. coli*, and *β-hemolytic Streptococcus* were present in 10–15% of cases. Some patients were also positive for *P. mirabilis*, *S. marcescens*, *C. freundi*, *P. stuartii*, *S. maltophilia*, *M. morganii*, and *Enterobacter*.

Erysipelas resembles cellulitis, but it usually affects the skin more superficially [20]. Patients present with erythematous edematous plaque with sudden onset, most often on the lower limbs (70–90% of cases), accompanied by unilateral inflammatory signs, chills, and regional adenopathy. Less frequent upper limb localization (5–10%) has been associated with lymphoedema after breast neoplasia in women. Facial lesions are the rarest (5% of cases). The main pathogens involved in erysipelas are *Streptococcus pyogenes* (58–67% of cases), *S. dysgalactiae* sp. *Echisimilis* (14–25%), and *S. agalactiae* (3–9%). Other bacteria found together with these streptococci are *Staphylococcus aureus* (10–17%), *Pseudomonas aeruginosa*, and enterobacteria (5–50%) [21]. In our study, in addition to these species, cultured samples also revealed colonization with *K. Pneumoniae* and *E. coli*, more so in the group with confirmed sepsis (17.9% of cases) than in the group with uncomplicated infections (8% *K. pneumoniae*, 11.8% *E. coli*).

Venous ulcers are a manifestation of chronic venous insufficiency (CVI), and they begin as small, round ulcerations that gradually expand to sometimes encompass the entire circumference of the calf. Edges are irregular, smooth, or slightly elevated, and the surface is covered with cellular debris from microbial colonization. Venous ulcers are typically located in the distal 1/3 of the calf, the internal malleolar region (60–70% of cases), less often on the calf’s middle third (18–20%), and most infrequently on the upper third (4–6%) or higher up the leg (3–4%) [22]. Venous ulcers were the most common pathology diagnosed in our patients (>60% of sepsis and uncomplicated infection cases), associated with bacterial superinfection with *S. aureus*, *P. aeruginosa*, *K. pneomoniae*, *E. coli*, as well as with para-venous microbial eczema.

Bullous dermatoses (pemphigus, bullous pemphigoid, erythema multiforme, Stevens–Johnson syndrome) can compromise the integrity of the epithelium and mucous membranes, thus undermining systemic homeostasis. Their etiopathogenic polymorphism is facilitated by numerous predisposing or favoring factors, to the detriment of good prognosis and quality of life [23]. In pemphigus, for instance, intraepidermal bullae are caused by the multifactorial activation of the immune system. The vesiculobullous lesions initially form in the mucous membranes and subsequently extend to the trunk. The buccal mucosa is damaged by flaccid vesicles that rupture easily and leave painful erosions, undermining the patient’s ability and willingness to eat, leading to weight loss and malnutrition. The cutaneous phase usually occurs after a latent period of several days to several months and consists of a bullous, monomorphic rash that spreads rapidly, sometimes over the entire body [24]. To achieve a good outcome, early bacteriological investigations, antibacterial therapy, and electrolyte and nutritional support are necessary [25]. Bullous pemphigoid is another such autoimmune condition that can progress to sepsis; in staphylococcal or streptococcal infection, autoantibodies can be triggered against multiprotein complexes that normally help basal epithelial cells adhere to the base membrane [26]. Even if less frequent, such cases constitute dermatological emergencies. In our study, aggravation to severe forms with septic complications and substantial hydro-electrolyte imbalance occurred in 3.8% of the 29 patients with superinfected bullous dermatoses.

As the name suggests, erythroderma is clinically expressed as redness (erythema), and it is diagnosed as such when more than 90% of the body surface is affected. Redness is not always a sign of infection; it can also occur in neoplasia or as hypersensitivity to medication. The estimated incidence is 1 in 100,000 adults, mostly men and typically aged 41–61, without racial predilection [27,28]. Typically, erythroderma is triggered by the exacerbation of a pre-existing dermatosis such as psoriasis (23% of cases), atopic dermatitis, and contact dermatitis [29]. Altered hemodynamic parameters, fever, hypoproteinemia/hypoalbuminemia, co-occurrence of edema, loss of fluids, electrolyte and/or acid-base imbalance, superinfection with risk of (or confirmed) sepsis justify taking this condition seriously [30]. Erythroderma was rare in our study (11 patients), but their clinical presentation data agrees with the literature.

Finally, a minority of patients also suffered from other dermatoses causing lesions disruptive of the skin barrier, e.g., psoriasis, atopic dermatitis, dermatitis herpetiformis, vasculitis, and ulcerated skin neoplasms. The absence from our analysis of conditions such as necrotizing fasciitis does not imply that such cases did not occur in our region during the studied period. Rather, our aims and methodology were not appropriate for the research of rare and severe skin infections that advance very quickly and require prompt surgical intervention.

### 4.3. The Role of Comorbidities in Acute Skin Infections and Related Sepsis

Comorbidities weaken the body’s defense mechanisms and can facilitate the spread of infection and eventually lead to sepsis by promoting the release of pro-inflammatory cytokines and altering coagulation processes, etc. In a previously cited Spanish study, 44% of patients hospitalized for soft tissue and skin infections had comorbidities; specifically, diabetes and heart disease were significant risk factors for major adverse events [15]. The patients included in our study presented a range of underlying conditions typical of the broader trends of illness in modern society. The multivariate analysis flagged type 1 diabetes mellitus and anemia as the most substantial risk factors for infection and sepsis, particularly in patients with cellulitis.

Hypertension was the most common comorbidity across all studied groups, especially in the sepsis group (69.2% vs. 58.8% of non-infected patients), but with non-significant differences. On the other hand, while 41% of patients with sepsis had established chronic heart failure, less than a quarter of patients with uncontaminated lesions did (23.5%). Importantly, sepsis is known to suppress myocardial function, favor diastolic dysfunction, and decrease the cardiac index and output. According to research data, patients with preserved ejection fraction are at a higher risk of mortality (2.4%) from sepsis than those without heart failure (0.4%), explained by insufficient cardiovascular reserves during systemic infection [31]. In our study, the infectious risk of patients was 1.924-fold higher in the presence of NYHA stage I and II heart failure.

During sepsis, the systemic release of pro-inflammatory cytokines, elevated levels of stress hormones, and changes in intravascular volume can compromise cardiovascular function, including electrical function. This can aggravate pre-existing arrhythmias or cause new-onset arrhythmias, particularly atrial fibrillation [32]. In our study, an established history of atrial fibrillation was significantly more frequent in the sepsis group (32.1%) compared to 27.8% of patients with uncomplicated infections and only 13.2% of non-infected patients. Established atrial fibrillation appeared to increase infectious risk by 2.527 times. Other published data show that sepsis is a risk factor for fibrillation, with new-onset atrial fibrillation being associated with a higher mortality rate than pre-existing fibrillation [33].

In our study, microcytic hypochromic anemia was present in 37.2% of patients with acute skin infections and 55.1% of patients with sepsis, suggesting that patients with anemia are twice as likely to develop sepsis. Similar results were obtained in a prospective study of ICU-admitted sepsis patients, where 42% had iron deficiency anemia. Iron is essential for host immunity and hemoglobin synthesis, which explains why patients with anemia are more vulnerable to sepsis [34]. In another study on 70 patients with sepsis and septic shock, as many as 82.6% had various forms of anemia related to chronic illness and hemolysis in severe stages of sepsis, recurring phlebotomy, and hemodilution. Hemolysis has been attributed to increased concentrations of hem, haptoglobin, hemopexin, and haemoxygenase-1 [35]. Although the cases of anemia in our study were limited in type to disturbances of systemic iron homeostasis, our results align with reports in the literature. Markers of hemolysis could be featured in future research of prognostic factors for cutaneous-onset sepsis.

Diabetes is a common underlying condition in patients who develop sepsis, yet sepsis prognosis in these patients is controversial [36,37]. Severe infections may be facilitated by inadequate glycemic control as insulin and oral antihyperglycemic drugs have been associated with lower incidence of sepsis. Diabetes is said to not increase the risk of mortality for sepsis patients, although it can promote kidney failure [38]. In both types 1 and 2, the pathological mechanisms contributing to higher risk of infection are complex and multifactorial. Glycemic imbalance can promote lesions via the irreversible glycation of protein chains, as well as by undermining the immune response and by generating oxygen-reactive species. Neuropathy and vascular lesions are diabetes-related complications that can further increase infection risk by interfering with leukocyte migration towards affected tissues, while peripheral artery disease reduces blood flow, limiting the effectiveness of antibiotic treatment and allowing bacteria to proliferate instead [37]. We found that as many as a quarter of patients with cutaneous-onset sepsis and a fifth of those with uncomplicated infections had type 2 diabetes mellitus. In addition, despite the narrowly defined nature of our study, patients with type 1 diabetes were at a particularly high (6.486-fold increased) risk of sepsis.

### 4.4. Further Comments on the Predictive Role of Mildly Altered Mental Status

Altered mental status is a major reason for concern in all medical specialties. It manifests as decreased cognitive function and level of awareness, confusion, behavioral changes, lack of alertness, and even coma. The Glasgow coma scores are often used to assess and monitor such types of neurological impairment [39]. Sometimes, the nervous system may be the first to show signs of dysfunction, particularly in elderly and immunocompromised patients, leading to a variety of clinical syndromes including sepsis-associated encephalopathy (SAE), seizures, stroke, and neuromuscular disorders [40].

Manifestations of sepsis-associated encephalopathy range from mild delirium to severe coma and are linked to increased long-term physical, mental, and cognitive dysfunction and increased mortality rates [41,42]. Reported incidence ranges from 9% to 71% [43,44,45]. Although SAE is a reversible syndrome, mild to moderate residual neuropsychiatric symptoms, including depression, anxiety, or cognitive impairment, may persist for up to one year in 40% of patients [46,47].

In the updated definition (sepsis-3), altered mental status expressed as Glasgow coma scores is an important predictor of sepsis. Its relevance has also been acknowledged for the Sequential Organ Failure Assessment (SOFA) score and the Quick SOFA version (qSOFA). These assess the degree of respiratory, renal, cardiovascular, neurological, hepatic, and hematological impairment, and qSOFA is used in the emergency department for faster organ failure scoring, taking into account respiratory rate, hypotension, and the Glasgow coma score [48,49]. Higher mortality rates have been reported in patients with GCSs <15, as well as neurological impairment in 25–33% of patients with sepsis [50,51].

In our study, mildly altered mental status was associated with a substantially higher risk of sepsis (5.866-fold increase), and the multivariate analysis also identified it as a significant predictor alongside typical systemic manifestations of infectious processes (fever, chills, adenopathy). More severely impaired patients would not have been referred from the emergency department to the dermatology unit. Nonetheless, our results suggest that dermatologists can be valuable contributors in cases involving cutaneous-onset sepsis.

### 4.5. Study Limitations and Further Research Opportunities

This study is intended as the first in a series for which our initial focus has been to assess the incidence of cutaneous-onset sepsis and infections in our region. The data hereby reported do not describe treatments and outcomes, which we could not collect in sufficient detail and with sufficient consistency from the type of electronic medical records to which we had access for the purposes of this study. The reported results provide a helpful basis for a prospective study (which we have already initiated) to assess specific biomarkers in patients who present to the emergency department with aggravated lesions. We are also in the process of reviewing diagnostic and therapeutic approaches separately to put forth a pilot algorithm for more prompt and efficient management of cutaneous-onset sepsis.

Readers should also be aware that, regarding the group of patients with uninfected lesions on admission, negative lab samples do not definitively rule out the possibility that these patients’ skin lesions had been infected previously. The opposite is suggested by the fact that these patients presented the clinical signs and symptoms typical of the studied conditions, which is why numbers for group C were not zero (e.g., the patient with clear signs of cellulitis but a negative bacteriological exam). It is common for patients to attempt to treat skin conditions at home using traditional antibiotic remedies, over the counter or prescription medication, ointments, etc., not always seeking or complying fully with professional advice from GPs or secondary care providers. This background information was not collected in this study, but it would be worth including it in future studies, whenever feasible.

Last but not least, this study did not and, indeed, could not have enrolled patients presenting in emergency departments with the primary skin lesions of interest as well as other complications or concomitant viral infections for which protocols required transfer to other units. Notably, the study period overlapped with the COVID-19 pandemic, so any patients carrying the SARS-CoV-2 virus were automatically directed to the hospital for infectious diseases.

## 5. Conclusions

Three years of admissions data at the largest emergency hospital in NE Romania were retrospectively assessed to establish the incidence of infections and sepsis relating specifically to primary skin lesions featuring loss of tissue. Over five hundred cases of adult patients were analyzed to identify which general characteristics, clinical manifestations, dermatological diagnoses, and established comorbidities were significant risk factors for infection and cutaneous-onset sepsis.

Patients presented ulcerations, erosions, fissures, excoriations, bullae, vesicles, pruritus, tumefaction, edema, fever, chills, pain, and adenopathy. They were diagnosed with venous ulcers, microbial eczema, cellulitis, superinfected bullous dermatoses, erysipelas, erythroderma, and other dermatoses. Of these, cellulitis was the strongest predictor of both infection and sepsis. While *S. aureus*, *P. aeruginosa*, and *E. coli* were the most common bacterial species evidenced in the cultured samples harvested from lesions on admission, *K. pneumoniae* and *S. β-hemolytic* were flagged as significant risk factors for infection. Mildly altered mental status was also found to be predictive of sepsis, especially in patients with type 1 diabetes mellitus and/or anemia. In fact, comorbidities were more strongly associated with the risk of infection (chronic heart failure, pre-existing atrial fibrillation) and sepsis (type 1 diabetes mellitus, anemia) than age, sex, or background, even if infections were more common in men and in rural residents.

These results contribute meaningfully to currently scarce research data on cutaneous-onset sepsis, indicating which primary skin lesions with loss of tissue can aggravate and become medical emergencies and which patients are most vulnerable. Further research ideas are put forth, including the study of biomarkers for cutaneous-onset sepsis that can be measured quickly and affordably, as well as the development of a pilot algorithm for effective management.

## Figures and Tables

**Figure 1 diagnostics-14-00659-f001:**
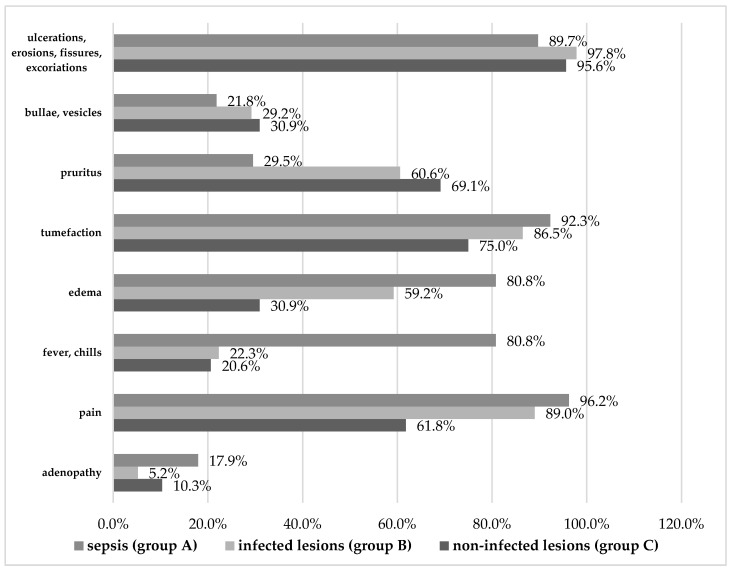
Clinical manifestations (skin-related and systemic) in the three study groups.

**Figure 2 diagnostics-14-00659-f002:**
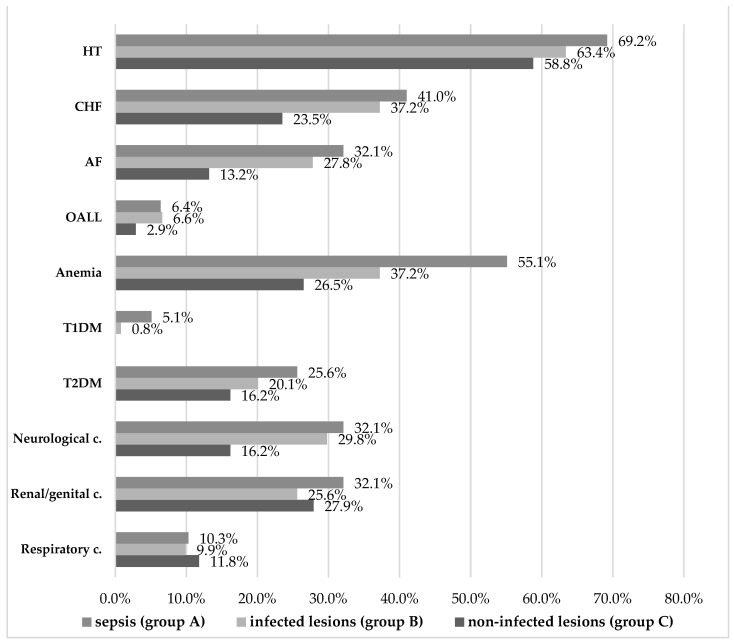
Comorbidities in the three study groups.

**Table 1 diagnostics-14-00659-t001:** Patient age data in the three study groups.

Study Group	N	Mean	St. Dev.	St. Error	Min	Max	Median
A—confirmed sepsis	78	61.33	15.022	1.701	18	88	62.0
B—infection without sepsis	363	65.23	14.056	0.738	18	92	65.0
C—non-infected lesions	68	62.13	17.075	2.071	20	90	65.5
Total	509	64.22	14.699	0.652	18	92	65.0

Infection present/absent: Mann–Whitney U test = 11,547.000; *p* = 0.399. Sepsis present/absent: Mann–Whitney U test = 12,122.000; *p* = 0.046.

**Table 2 diagnostics-14-00659-t002:** Demographic characteristics—group A vs. group B.

	Total		Sepsis				Pearson Chi-Squared		OR 95% CI
			Yes (Group A)		No (Group B)				
	N	%	N	%	N	%	Chi^2^	*p*	
Sex							0.519	0.471	-
male	248	56.2%	41	52.6%	207	57.0%			
female	193	43.8%	37	47.4%	156	43.0%			
Background							0.481	0.488	-
urban	188	42.6%	36	46.2%	152	41.9%			
rural	253	57.4%	42	53.8%	211	58.1%			
Total	441	100%	78	100%	363	100%			

**Table 3 diagnostics-14-00659-t003:** Clinical manifestations in patients with infected lesions (with vs. without sepsis).

	Total		Sepsis				Pearson Chi-Squared		OR 95% CI
			Yes (Group A)		No (Group B)				
	N	%	N	%	N	%	Chi^2^	*p*	
Ulcerations, erosions, fissures, excoriations							11.907	0.003 *	0.197 (0.072 ÷ 0.543)
present	425	96.4%	70	89.7%	355	97.8%			
absent	16	3.6%	8	10.3%	8	2.2%			
Bullae, vesicles							1.751	0.186	-
present	123	27.9%	17	21.8%	106	29.2%			
absent	318	72.1%	61	78.2%	257	70.8%			
Pruritus							25.131	<0.001 *	0.272 (0.160 ÷ 0.462)
present	243	55.1%	23	29.5%	220	60.6%			
absent	198	44.9%	55	70.5%	143	39.4%			
Tumefaction							1.983	0.159	-
present	386	87.5%	72	92.3%	314	86.5%			
absent	55	12.5%	6	7.7%	49	13.5%			
Edema							12.786	<0.001 **	2.891 (1.586 ÷ 5.272)
present	278	63.0%	63	80.8%	215	59.2%			
absent	163	37.0%	15	19.2%	148	40.8%			
Fever, chills							99.762	<0.001 **	14.622 (7.906 ÷ 27.044)
present	144	32.7%	63	80.8%	81	22.3%			
absent	297	67.3%	15	19.2%	282	77.7%			
Pain							3.754	0.053	-
present	398	90.2%	75	96.2%	323	89.0%			
absent	43	9.8%	3	3.8%	40	11.0%			
Adenopathy							14.992	<0.001 **	3.961 (1.889 ÷ 8.302)
present	33	7.5%	14	17.9%	19	5.2%			
absent	408	92.5%	64	82.1%	344	94.8%			
Total	441	100%	78	100%	363	100%			

* statistically significant results (*p* < 0.05); ** strong statistical significance (*p* < 0.001).

**Table 4 diagnostics-14-00659-t004:** Clinical manifestations in patients with infected vs. non-infected lesions.

	Total		Infection				Pearson Chi-Squared		OR 95% CI
			Yes (Group B)		No (Group C)				
	N	%	N	%	N	%	Chi^2^	*p*	
Ulcerations, erosions, fissures, excoriations							1.123	0.391	-
present	420	97.4%	355	97.8%	65	95.6%			
absent	11	2.6%	8	2.2%	3	4.4%			
Bullae, vesicles							0.078	0.780	-
present	127	29.5%	106	29.2%	21	30.9%			
absent	304	70.5%	257	70.8%	47	69.1%			
Pruritus							1.760	0.185	-
present	267	61.9%	220	60.6%	47	69.1%			
absent	164	38.1%	143	39.4%	21	30.9%			
Tumefaction							5.842	0.016 *	2.136 (1.142 ÷ 3.995)
present	365	84.7%	314	86.5%	51	75%			
absent	66	15.3%	49	13.5%	17	25%			
Edema							18.575	<0.001 **	3.251 (1.866 ÷ 5.666)
present	236	54.8%	215	59.2%	21	30.9%			
absent	195	45.2%	148	40.8%	47	69.1%			
Fever, chills							0.099	0.753	-
present	95	22%	81	22.3%	14	20.6%			
absent	336	78%	282	77.7%	54	79.4%			
Pain							32.712	<0.001 **	4.999 (2.773 ÷ 9.010)
present	365	84.7%	323	89%	42	61.8%			
absent	66	15.3%	40	11%	26	38.2%			
Adenopathy							2.587	0.158	-
present	26	6%	19	5.2%	7	10.3%			
absent	405	94%	344	94.8%	61	89.7%			
Total	431	100%	363	100%	68	100%			

* statistically significant results (*p* < 0.05); ** strong statistical significance (*p* < 0.001).

**Table 5 diagnostics-14-00659-t005:** Glasgow scores—patient group A vs. group B.

Glasgow Score 13–14	Total		Sepsis				Pearson Chi-Squared		OR 95% CI
			Yes (Group A)		No (Group B)				
	N	%	N	%	N	%	Chi^2^	*p*	
present	13	2.9%	7	9%	6	1.7%	12.030	0.003 *	5.866 (1.914 ÷ 17.975)
absent	428	97.1%	71	91%	357	98.3%			
Total	441	100%	78	100%	363	100%			

* statistically significant results (*p* < 0.05).

**Table 6 diagnostics-14-00659-t006:** Acute skin conditions in patients with vs. without sepsis.

	Total		Sepsis				Pearson Chi-Squared		OR 95% CI
			Yes (Group A)		No (Group B)				
	N	%	N	%	N	%	Chi^2^	*p*	
Venous ulcers							0.110	0.740	-
present	301	68.3%	52	66.7%	249	68.6%			
absent	140	31.7%	26	33.3%	114	31.4%			
Microbial eczema							42.227	<0.001 **	0.189 (0.111 ÷ 0.323)
present	273	61.9%	23	29.5%	250	68.9%			
absent	168	38.1%	55	70.5%	113	31.1%			
Cellulitis							43.470	<0.001 **	5.320 (3.136 ÷ 9.026)
present	93	21.1%	38	48.7%	55	15.2%			
absent	348	78.9%	40	51.3%	308	84.8%			
Superinfected bullous dermatoses							1.149	0.284	-
present	29	6.6%	3	3.8%	26	7.2%			
absent	412	93.4%	75	96.2%	337	92.8%			
Erysipelas							5.548	0.038 *	2.849 (1.152 ÷ 7.048)
present	22	5%	8	10.3%	14	3.9%			
absent	419	95%	70	89.7%	349	96.1%			
Erythroderma							4.518	0.056	3.870 (1.015 ÷ 14.757)
present	9	2%	4	5.1%	5	1.4%			
absent	432	98%	74	94.9%	358	98.6%			
Other dermatoses featuring loss of tissue or the disruption of the skin barrier							2.801	0.094	-
present	119	27%	27	34.6%	92	25.3%			
absent	322	73%	51	65.4%	271	74.7%			
Total	441	100%	78	100%	363	100%			

* statistically significant results (*p* < 0.05); ** strong statistical significance (*p* < 0.001).

**Table 7 diagnostics-14-00659-t007:** Acute skin conditions in patients with uncomplicated infections vs. without infection.

	Total		Infection				Pearson Chi-Squared		OR 95% CI
			Yes (Group B)		No (Group C)				
	N	%	N	%	N	%	Chi^2^	*p*	
Venous ulcers							73.049	<0.001 **	14.319 (6.862 ÷ 29.878)
present	258	59.9%	249	68.6%	9	13.2%			
absent	173	40.1%	114	31.4%	59	86.8%			
Microbial eczema							40.897	<0.001 **	5.706 (3.213 ÷ 10.134)
present	269	62.4%	250	68.9%	19	27.9%			
absent	162	37.6%	113	31.1%	49	72.1%			
Cellulitis							9.482	0.002 *	11.964 (1.627 ÷ 87.988)
present	56	13.0%	55	15.2%	1	1.5%			
absent	375	87.0%	308	84.8%	67	98.5%			
Superinfected bullous dermatoses							0.230	0.632	-
present	32	7.4%	26	7.2%	6	8.8%			
absent	399	92.6%	337	92.8%	62	91.2%			
Erysipelas							19.214	<0.001 **	0.187 (0.082 ÷ 0.425)
present	26	6.0%	14	3.9%	12	17.6%			
absent	405	94.0%	349	96.1%	56	82.4%			
Erythroderma							0.877	0.305	-
present	7	1.6%	5	1.4%	2	2.9%			
absent	424	98.4%	358	98.6%	66	97.1%			
Other dermatoses featuring loss of tissue or the disruption of the skin barrier							27.735	<0.001 **	0.252 (0.148 ÷ 0.431)
present	131	30.4%	92	25.3%	39	57.4%			
absent	300	69.6%	271	74.7%	29	42.6%			
Total	431	100%	363	100%	68	100%			

* statistically significant results (*p* < 0.05); ** strong statistical significance (*p* < 0.001).

**Table 8 diagnostics-14-00659-t008:** Etiological agents in the infected lesions (with vs. without sepsis).

	Total		Sepsis				Pearson Chi-Squared		OR 95% CI
			Yes (Group A)		No (Group B)				
	N	%	N	%	N	%	Chi^2^	*p*	
*S. aureus*							1.451	0.228	-
present	191	43.3%	29	37.2%	162	44.6%			
absent	250	56.7%	49	62.8%	201	55.4%			
*P. aeruginosa*							0.029	0.865	-
present	149	33.8%	27	34.6%	122	33.6%			
absent	292	66.2%	51	65.4%	241	66.4%			
*E. coli*							2.125	0.145	-
present	57	12.9%	14	17.9%	43	11.8%			
absent	384	87.1%	64	82.1%	320	88.2%			
*K. pneumoniae*							7.237	0.007 *	2.519 (1.262 ÷ 5.031)
present	43	9.8%	14	17.9%	29	8%			
absent	398	90.2%	64	82.1%	334	92%			
*S. β-hemolytic*							5.438	0.031 *	2.655 (1.137 ÷ 6.200)
present	26	5.9%	9	11.5%	17	4.7%			
absent	415	94.1%	69	88.5%	346	95.3%			
Other bacterial species							0.124	0.724	-
present	134	30.4%	25	32.1%	109	30%			
absent	307	69.6%	53	67.9%	254	70%			
Viral agents							0.033	0.855	-
present	42	9.5%	7	9%	35	9.6%			
absent	399	90.5%	71	91%	328	90.4%			
Total	441	100%	78	100%	363	100%			

* statistically significant results (*p* < 0.05).

**Table 9 diagnostics-14-00659-t009:** Comorbidities in patients with infected lesions—with vs. without sepsis.

	Total		Sepsis				Pearson Chi-Squared		OR 95% CI
			Yes (Group A)		No (Group B)				
	N	%	N	%	N	%	Chi^2^	*p*	
Hypertension							0.965	0.326	-
present	284	64.4%	54	69.2%	230	63.4%			
absent	157	35.6%	24	30.8%	133	36.6%			
Chronic heart failure							0.401	0.526	-
present	167	37.9%	32	41%	135	37.2%			
absent	274	62.1%	46	59%	228	62.8%			
Atrial fibrillation							0.562	0.453	-
present	126	28.6%	25	32.1%	101	27.8%			
absent	315	71.4%	53	67.9%	262	72.2%			
Obliterating arteriopathy of the lower limbs (OALL)							0.004	0.948	-
present	29	6.6%	5	6.4%	24	6.6%			
absent	412	93.4%	73	93.6%	339	93.4%			
Anemia							8.583	0.003 *	2.075 (1.266 ÷ 3.402)
present	178	40.4%	43	55.1%	135	37.2%			
absent	263	59.6%	35	44.9%	228	62.8%			
Type 1 diabetes mellitus							7.606	0.021 *	6.486 (1.422 ÷ 29.590)
present	7	1.6%	4	5.1%	3	0.8%			
absent	434	98.4%	74	94.9%	360	99.2%			
Type 2 diabetes mellitus							1.180	0.277	-
present	93	21.1%	20	25.6%	73	20.1%			
absent	348	78.9%	58	74.4%	290	79.9%			
Neurological conditions							0.161	0.688	-
present	133	30.2%	25	32.1%	108	29.8%			
absent	308	69.8%	53	67.9%	255	70.2%			
Renal/genital conditions							1.355	0.244	-
present	118	26.8%	25	32.1%	93	25.6%			
absent	323	73.2%	53	67.9%	270	74.4%			
Respiratory conditions							0.008	0.928	-
present	44	10%	8	10.3%	36	9.9%			
absent	397	90%	70	89.7%	327	90.1%			
Total	441	100%	78	100%	363	100%			

* statistically significant results (*p* < 0.05).

**Table 10 diagnostics-14-00659-t010:** Comorbidities in patients with infected lesions (without sepsis) vs. non-infected lesions.

	Total		Infection				Pearson Chi-Squared		OR 95% CI
			Yes (Group B)		No (Group C)				
	N	%	N	%	N	%	Chi^2^	*p*	
Hypertension							0.504	0.478	-
present	270	62.6%	230	63.4%	40	58.8%			
absent	161	37.4%	133	36.6%	28	41.2%			
Chronic heart failure							4.696	0.030 *	1.924 (1.057 ÷ 3.504)
present	151	35.0%	135	37.2%	16	23.5%			
absent	280	65.0%	228	62.8%	52	76.5%			
Atrial fibrillation							6.412	0.011 *	2.527 (1.208 ÷ 5.286)
present	110	25.5%	101	27.8%	9	13.2%			
absent	321	74.5%	262	72.2%	59	86.8%			
Obliterating arteriopathy of the lower limbs (OALL)							1.361	0.402	-
present	26	6.0%	24	6.6%	2	2.9%			
absent	405	94.0%	339	93.4%	66	97.1%			
Anemia							2.874	0.090	-
present	153	35.5%	135	37.2%	18	26.5%			
absent	278	64.5%	228	62.8%	50	73.5%			
Type 1 diabetes mellitus							0.566	1.000	-
present	3	0.7%	3	0.8%		0%			
absent	428	99.3%	360	99.2%	68	100%			
Type 2 diabetes mellitus							0.565	0.452	-
present	84	19.5%	73	20.1%	11	16.2%			
absent	347	80.5%	290	79.9%	57	83.8%			
Neurological conditions							5.281	0.022 *	2.195 (1.108 ÷ 4.347)
present	119	27.6%	108	29.8%	11	16.2%			
absent	312	72.4%	255	70.2%	57	83.8%			
Renal/genital conditions							0.160	0.689	-
present	112	26.0%	93	25.6%	19	27.9%			
absent	319	74.0%	270	74.4%	49	72.1%			
Respiratory conditions							0.213	0.644	-
present	44	10.2%	36	9.9%	8	11.8%			
absent	387	89.8%	327	90.1%	60	88.2%			
Total	431	100%	363	100%	68	100%			

* statistically significant results (*p* < 0.05).

**Table 11 diagnostics-14-00659-t011:** Multivariate analysis of risk factors for infection.

Risk Factors				95% CI	
(Statistically Significant)	Coef. B	*p*-Value	Odds Ratio	Lower	Upper
Rural background	1.066	0.001 *	2.903	1.521	5.538
Neurological conditions	1.203	0.002 *	3.329	1.527	7.260
*E. coli*	2.368	0.026 *	10.679	1.335	85.416
Venous ulcers	2.205	<0.001 **	9.073	3.833	21.480
Cellulitis	2.205	0.037 *	9.071	1.137	72.347
Microbial eczema	1.130	0.002 *	3.094	1.489	6.432
Constant	−0.789	0.009	0.454		

* statistically significant results (*p* < 0.05); ** strong statistical significance (*p* < 0.001).

**Table 12 diagnostics-14-00659-t012:** Risk of infection in the combined presence of individual risk factors.

Rural Residence	Neurological Conditions	*E. coli*	Venous Ulcers	Cellulitis	Microbial Eczema	Risk of Infection
0	0	0	0	0	0	0.31
0	0	0	0	0	1	0.58
0	0	0	0	1	0	0.80
0	0	0	0	1	1	0.93
0	0	0	1	0	0	0.80
0	0	0	1	0	1	0.93
0	0	0	1	1	0	0.97
0	0	0	1	1	1	0.99
0	0	1	0	0	0	0.83
0	0	1	0	0	1	0.94
0	0	1	0	1	0	0.98
0	0	1	0	1	1	0.99
0	0	1	1	0	0	0.98
0	0	1	1	0	1	0.99
0	0	1	1	1	0	1.00
0	0	1	1	1	1	1.00
0	1	0	0	0	0	0.60
0	1	0	0	0	1	0.82
0	1	0	0	1	0	0.93
0	1	0	0	1	1	0.98
0	1	0	1	0	0	0.93
0	1	0	1	0	1	0.98
0	1	0	1	1	0	0.99
0	1	0	1	1	1	1.00
0	1	1	0	0	0	0.94
0	1	1	0	0	1	0.98
0	1	1	0	1	0	0.99
0	1	1	0	1	1	1.00
0	1	1	1	0	0	0.99
0	1	1	1	0	1	1.00
0	1	1	1	1	0	1.00
0	1	1	1	1	1	1.00
1	0	0	0	0	0	0.57
1	0	0	0	0	1	0.80
1	0	0	0	1	0	0.92
1	0	0	0	1	1	0.97
1	0	0	1	0	0	0.92
1	0	0	1	0	1	0.97
1	0	0	1	1	0	0.99
1	0	0	1	1	1	1.00
1	0	1	0	0	0	0.93
1	0	1	0	0	1	0.98
1	0	1	0	1	0	0.99
1	0	1	0	1	1	1.00
1	0	1	1	0	0	0.99
1	0	1	1	0	1	1.00
1	0	1	1	1	0	1.00
1	0	1	1	1	1	1.00
1	1	0	0	0	0	0.81
1	1	0	0	0	1	0.93
1	1	0	0	1	0	0.98
1	1	0	0	1	1	0.99
1	1	0	1	0	0	0.98
1	1	0	1	0	1	0.99
1	1	0	1	1	0	1.00
1	1	0	1	1	1	1.00
1	1	1	0	0	0	0.98
1	1	1	0	0	1	0.99
1	1	1	0	1	0	1.00
1	1	1	0	1	1	1.00
1	1	1	1	0	0	1.00
1	1	1	1	0	1	1.00
1	1	1	1	1	0	1.00
1	1	1	1	1	1	1.00

**Table 13 diagnostics-14-00659-t013:** Multivariate analysis of risk factors for sepsis.

Risk Factors				95% CI	
(Statistically Significant)	Coef. B	*p*-Value	Odds Ratio	Lower	Upper
Glasgow score 13, 14	1.696	0.023 *	5.451	1.268	23.432
Fever, chills	2.251	<0.001 **	9.497	4.703	19.177
Adenopathy	1.384	0.003 *	3.993	1.594	10.003
Type 1 diabetes mellitus	2.668	0.005 *	14.404	2.205	94.101
Anemia	0.678	0.026 *	1.971	1.086	3.578
Cellulitis	1.180	<0.001 **	3.253	1.652	6.404
Constant	−3.703	<0.001	0.025		

* statistically significant results (*p* < 0.05); ** strong statistical significance (*p* < 0.001).

**Table 14 diagnostics-14-00659-t014:** Risk of sepsis in the combined presence of individual risk factors.

Glasgow Scores 13, 14	Fever, Chills	Adenopathy	T1DM	Anemia	Cellulitis	Risk of Sepsis
0	0	0	0	0	0	0.02
0	0	0	0	0	1	0.07
0	0	0	0	1	0	0.05
0	0	0	0	1	1	0.14
0	0	0	1	0	0	0.26
0	0	0	1	0	1	0.54
0	0	0	1	1	0	0.41
0	0	0	1	1	1	0.69
0	0	1	0	0	0	0.09
0	0	1	0	0	1	0.24
0	0	1	0	1	0	0.16
0	0	1	0	1	1	0.39
0	0	1	1	0	0	0.59
0	0	1	1	0	1	0.82
0	0	1	1	1	0	0.74
0	0	1	1	1	1	0.90
0	1	0	0	0	0	0.19
0	1	0	0	0	1	0.43
0	1	0	0	1	0	0.32
0	1	0	0	1	1	0.60
0	1	0	1	0	0	0.77
0	1	0	1	0	1	0.92
0	1	0	1	1	0	0.87
0	1	0	1	1	1	0.96
0	1	1	0	0	0	0.48
0	1	1	0	0	1	0.75
0	1	1	0	1	0	0.65
0	1	1	0	1	1	0.86
0	1	1	1	0	0	0.93
0	1	1	1	0	1	0.98
0	1	1	1	1	0	0.96
0	1	1	1	1	1	0.99
1	0	0	0	0	0	0.12
1	0	0	0	0	1	0.30
1	0	0	0	1	0	0.21
1	0	0	0	1	1	0.46
1	0	0	1	0	0	0.66
1	0	0	1	0	1	0.86
1	0	0	1	1	0	0.79
1	0	0	1	1	1	0.93
1	0	1	0	0	0	0.35
1	0	1	0	0	1	0.64
1	0	1	0	1	0	0.51
1	0	1	0	1	1	0.77
1	0	1	1	0	0	0.89
1	0	1	1	0	1	0.96
1	0	1	1	1	0	0.94
1	0	1	1	1	1	0.98
1	1	0	0	0	0	0.56
1	1	0	0	0	1	0.81
1	1	0	0	1	0	0.72
1	1	0	0	1	1	0.89
1	1	0	1	0	0	0.95
1	1	0	1	0	1	0.98
1	1	0	1	1	0	0.97
1	1	0	1	1	1	0.99
1	1	1	0	0	0	0.84
1	1	1	0	0	1	0.94
1	1	1	0	1	0	0.91
1	1	1	0	1	1	0.97
1	1	1	1	0	0	0.99
1	1	1	1	0	1	1.00
1	1	1	1	1	0	0.99
1	1	1	1	1	1	1.00

## Data Availability

The data presented in this study are available on request from the corresponding author.

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
