# Peer review of "Skin Lesions with Loss of Tissue and Cutaneous-Onset Sepsis: The Skin Infection–Sepsis Relationship"

_diagnostics, 2024, doi:10.3390/diagnostics14060659_

Round 1

Reviewer 1 Report

Comments and Suggestions for Authors

Comments on the Quality of English Language

English is acceptable but the text must be checked for typos and formal errors

Author Response

Thank you.

Reviewer 2 Report

Comments and Suggestions for Authors

Skin problems involving tissue loss, like ulcers and blisters, can sometimes lead to infections and even life-threatening sepsis. This retrospective study, conducted in emergency hospital in northeastern Romania, looked at data from patients admitted between 2020 and 2022 on patients between 18 and 92 years old who presented with various skin issues like ulcers, redness, cracks and blisters. Using patient specific symptoms the authors draw correlative conclusions between specific types of skin conditions/systemic status of the patient/other comorbidities and sepsis occurrence with significant hazard ratios. Firstly, the authors found that over 441 out of 509 patients had infected skin lesions, and 78 developed sepsis. Sepsis development was found to be likely caused by venous ulcers, cellulitis and ethyroderma. Additionally, patients with existing health problems, such as diabetes, heart failure and anemia were found to be at a higher risk for both infection and sepsis. Further microbiome analysis of infected lesions found prominent presence of Staphylococcus aureus, Pseudomonas aeruginosa, and Escherichia coli. Correlative conclusions have also be drawn with respect to neurological conditions (mild deterioration of mental status assessed using Glasgow Coma scale) and sepsis occurrence. Another interesting observations the authors have derived at is the correlation between men and people living rural areas being prone to develop infections compared to women and people living in urban areas. Overall, these findings highlight the importance of further research in this area to develop better ways to diagnose and manage skin infections and prevent sepsis, especially in patients with weakened immune systems. The sample size taken is significant enough to draw valid and relevant conclusions.

This can be accepted for publication pending minor grammatical errors. Primarily, in several sentences, spacing between two words appears to be an issue (line nos: 98, 108, 134, 152, 199, 233, 281, 290, 322, 412, 421, 423, 424, 433, 493, 505, 567, 569, 617, 631, 635, 636) 645,

The abbreviation NYHA has to be expanded at first use.

The methods section lack information on how the microbial species were detected in infected lesions. This can be provided.

Comments on the Quality of English Language

Minor grammatical errors noted and needs to be improved.

Author Response

Thank you.
